# Preparation of MCS from Low-Grade Bauxite Desilication Lye and Adsorption of Heavy Metals

**DOI:** 10.3390/ma16093506

**Published:** 2023-05-02

**Authors:** Cheng Chen, Chaoyi Chen, Junqi Li, Gangan Wang, Xin Lin, Deyang Ning

**Affiliations:** 1College of Materials and Metallurgy, Guizhou University, Guiyang 550025, China; 2Guizhou Province Key Laboratory of Metallurgical Engineering and Process Energy Saving, Guiyang 550025, China

**Keywords:** calcium mesoporous silicate, heavy metal, low grade bauxite, adsorption

## Abstract

By utilizing low-grade bauxite desilication solution as raw material and adding lime after thermal reaction, adsorbent MCS was synthesized. X-ray diffraction, Brunauer–Emmett–Teller, Fourier transform infrared spectroscopy, and scanning electron microscopy were used to characterize the MCS, MCS-Pb, and MCS-Cu. The Freundlich model was found to be more suitable for isothermal adsorption, suggesting that the adsorption of Cu^2+^ and Pb^2+^ by MCS is not limited to monolayer adsorption. According to the results of the experiment, the maximum adsorption capacities of lead ion and copper ion were found to be Pb^2+^ (1921.506 mg/g) > Cu^2+^ (561.885 mg/g), and the adsorption was controlled by chemical reactions following pseudo-second-order kinetics. Electrolyte study results indicated that the presence of background electrolyte did not affect the adsorption of Cu^2+^ and Pb^2+^ by MCS.

## 1. Introduction

The main sources of heavy metal ions in industrial wastewater are smelting engineering, electroplating, and electrolytic processes. Water containing heavy metals is difficult to biodegrade, easily accumulates, and can cause serious toxicity reactions [1]. Therefore, the effective removal of heavy metal ions from water has become an urgent issue in the present day. Meanwhile, researchers have conducted extensive studies on the adsorption of heavy metal ions, resulting in an increasing interest in determining how to solve this problem efficiently and economically.

There are many traditional methods for treating heavy metal wastewater, including chemical precipitation, chelate flocculation, ferrite, ion exchange, membrane separation, electrochemistry, flotation, and adsorption. Due to its low cost and environmental friendliness, adsorption is one of the most used methods of removing heavy metals from solutions. Abukhadra [2] prepared kaolinite nanotubes for heavy metal adsorption using kaolinite as a precursor and demonstrated that the process not only involved chemical exchange and ion exchange, but also multilayer adsorption. Furthermore, the adsorbent is reusability-friendly and can be used in the future. Sharma [3] synthesized mesoporous zinc oxide and titanium dioxide using a nanoporous casting technique. These adsorbents are effective at adsorbing Pb^2+^ and Cd^2+^ from the solution. Ghiloufi [4] also obtained similar results. According to the Italian wastewater regulations, Di [5] used silica-supported hydrophilic carbon nanoparticles to adsorb heavy metals Cb^2+^, Ni^2+^, and Pb^2+^ in the solution with high adsorption rates. Li [6] prepared a polymer composite by utilizing a variety of solid wastes for the adsorption of heavy metals Pb^2+^ and Cd^2+^. During the synthesis of the composite, calcium silicate was formed and contributed to improved mechanical properties. Czech et al. [7] used calcium silicate for the adsorption of Cd^2+^ heavy metal, which provided advantages such as good adsorption capacity, a broad range of raw materials, and economic feasibility. A significant improvement in the phosphorus removal rate was observed when free Ca^2+^ on the surface of calcium silicate was washed away by deionized water in a study by Okano et al. [8]. Salem et al. [9] used nanoscale ZnO as an adsorbent for heavy metal ions, which was later used to adsorb acidic dye Acid Blue 92 (AB92) in aqueous solutions. Shao et al. [10] synthesized calcium silicate for heavy metal adsorption, which was subsequently used in the degradation of MB as a photocatalyst. As a result of this experiment path, solid waste adsorbents are utilized for heavy metal adsorption, followed by their use as catalysts. This provides a new reference for the treatment of adsorbents.

As a summary, synthesis of mesoporous calcium silicate (MCS) from low-quality bauxite ore using desilication liquid can effectively and economically remove silica and regenerate the alkali solution used in desilication. The MCS has excellent performance in adsorbing metal ions from water and is widely available, easily synthesized, inexpensive, and has many potential applications.

## 2. Experimental

### 2.1. Materials

Low-grade bauxite and limestone were obtained from an alumina plant in Guizhou. The limestone was calcined at 1000 °C for 3 h prior to use to increase its reactivity (effective CaO content was approximately 80%). Nitric acid (HNO_3_, purity ≥ 99 %), copper nitrate trihydrate (Cu(NO_3_)_2_ 3H_2_O, purity ≥ 99%), lead nitrate hexahydrate (Pb(NO_3_)_2_·6H_2_O, purity ≥ 99%), sodium nitrate (NaNO_3_, purity ≥ 98%), and sodium hydroxide (NaOH, purity ≥ 98%) were purchased from Aladdin Chemistry Co. Ltd. (Shanghai, China). Deionized water was prepared in-house at Guizhou University, and its conductivity was less than 1 μS/cm.

### 2.2. Synthesis of MCS

Initially, low-grade bauxite was reacted with 110 g/L caustic soda solution at a liquid-to-solid ratio of 10 under 95 °C for 30 min, and a silicon-containing alkali solution was obtained. After that, MCS was prepared by hydrothermal reaction of silicon-containing alkali solution with lime-to-calcium-silicon ratio of 2 at 95 °C for 120 min, followed by filtration, washing, and drying.

The XRD and SEM patterns of the prepared MCS are shown in Figures 2 and 4a. It can be seen from Figure 2 that the peak of MCS was narrow, indicating that its grain size was significant, and it mainly consisted of calcium silicate gel, calcium hydroxide Ca(OH)_2_, and other phases. The porous structure of MCS is shown in Figure 4a, which can provide more adsorption sites for the adsorption process.

### 2.3. Analytical Method

The overall surface morphology was investigated using a BrukerAXS D8 Advance X-ray diffractometer, a scanning electron microscope, and energy dispersive X-ray spectroscopy (TESCAN MIRA LMS, Brno, Czech Republic). Brunauer–Emmett–Teller (BET) surface analysis was carried out using a Micromeritics Tristar II 3020 instrument (Mac ASAP2460, MICROMERITICS, Atlanta, GA, USA), while changes in the chemical bonds of MCS were identified using Fourier transform infrared spectroscopy (FT-IR) (Nicolet 670, Nicolite, Madison, WI, USA). A spectrophotometer (PERSEE A3G, Beijing, China) was used to detect heavy metal ions in the solution, and a pH meter (PHS = 3E, Shanghai, China) was used to monitor pH changes.

### 2.4. Adsorption Experiment

MCS was used to adsorb copper or lead ions. The experimental process was as follows: different NaOH and HNO_3_ solution concentrations were used to adjust the pH of Cu^2+^ and Pb^2+^ solutions with 200 mg/L concentrations. Then, 0.0040 g, 0.0080 g, 0.0120 g, 0.0160 g, 0.0240 g of MCS and 20 mL of Cu^2+^ and Pb^2+^ solutions were added at a concentration of 200 mg/L, pH = 2.00~6.00, respectively, to a 100 mL centrifuge tube. After 2 min of ultrasound with a high-power CNC ultrasonic cleaner, the oscillations were carried out in a water bath thermostatic oscillator at 303 K for 24 h. As a result of the shaking, the samples were collected and then filtered through a 0.22 m pinhole filter using a 5 mL syringe. Through the use of a stoppered colorimetric tube, the diluted samples were volumetrically determined using atomic absorption spectrometry, and then the Cu^2+^ and Pb^2+^ concentrations were determined using atomic absorption spectrometry to determine the removal rates and adsorption capacities of Cu^2+^ and Pb^2+^.

The determination results were substituted into (1) and (2) to calculate the adsorption capacity and removal rate. The adsorption capacity (q, mg∙g^−1^) and removal efficiency (R, %) of the adsorbent were calculated by the following formula:(1)$$q=\frac{\left(C_{0}-C_{e}\right)v}{m}$$
(2)$$R=\frac{C_{0}-C_{e}}{C_{0}}\times 100\;\%$$

C_0_ and C_e_ represent the concentration of adsorbed substance under initial and equilibrium states, respectively, mg/L; m represents the mass of the adsorbent, g; v is the volume of the adsorbent solution, L.

### 2.5. Isothermal Adsorption Model

The adsorption isotherm is a graph plotting the amount of adsorbent removed from the system solution against the amount remaining in the solution at isothermal conditions. The adsorption isotherm allows for the study of the proportion of adsorbent molecules distributed between the solid and liquid phases at equilibrium [11,12]. 

#### 2.5.1. Langmuir Model

According to the law of mass action and the characteristic of equal adsorption and desorption rates at adsorption equilibrium, the isothermal equation for adsorption is derived as [11,12]:(3)$$\frac{C_{e}}{q_{e}}=\frac{1}{q_{m}K_{L}}+\frac{C_{e}}{q_{m}}$$
where C_e_: liquid phase concentration at adsorption equilibrium, mg/L; C_0_: initial concentration of liquid phase, mg/L; q_e_: adsorption capacity at equilibrium, mg/g; K_L_: Langmuir adsorption isothermal equation coefficient, L/mg; q_m_: maximum adsorption capacity of adsorbent per unit mass, mg/g; R_L_: Langmuir adsorption equilibrium constant.

Using C_e_/q_e_ as a graph of C_e_, the adsorption capacity of q_max_ and K_L_ is generally related to the specific surface area and porosity of the adsorbent based on the slope and intercept of the line. When the adsorbent has a high specific surface area and pore, it generally has a high adsorption capacity [13]. Langmuir’s adsorption isothermal model is suitable for uniform adsorption, in which each molecule on the surface of the adsorbent has the same adsorption activation energy [14].

Adsorption that conforms to the Langmuir isotherm is chemisorption. It is generally believed that the activation energy of chemisorption is in the range of 40–400 kJ/mol and that, except in special circumstances, a spontaneous chemisorption process, which should be exothermic, will decrease in saturation adsorption with increasing isotherm [15]. K_L_ is the equilibrium constant for adsorption, also known as the adsorption coefficient. Its magnitude is related to the adsorbent, the nature of the adsorbent, and the temperature; the larger the K_L_ value, the stronger the adsorption capacity, and K_L_ has the magnitude of the inverse of the concentration [16,17].

#### 2.5.2. Freundlich Model

The Freundlich adsorption isotherm describes the heterogeneous system, namely reversible adsorption. This isotherm is not limited to single-layer adsorption, and its equation is [18,19]:*q_e_* = K_F_C_e_^1/*n*^(4)
where *q_e_*: the mass of adsorbent adsorbed by a unit adsorbent at equilibrium, mg/g; C_e_: adsorption mass equilibrium concentration, mg/L; K_F_: Freundlich adsorption isothermal equation equilibrium constant, L/g; 1/*n*: Freundlich adsorption isothermal equation equilibrium constant.

It is generally believed that the value of 1/*n* is between 0 and 1, and its value indicates the strength of the influence of concentration on adsorption capacity. The smaller the 1/*n*, the better the adsorption performance [20]. Therefore, 1/*n* in 0.1~0.5, that is, *n* between 2~10, it is easy to adsorb; when 1/*n* is greater than 2, that is, *n* is between 0 and 2, it is difficult to adsorb [21]. 

### 2.6. Adsorption Kinetics

Models of adsorption kinetics are used in order to study the mechanism of diffusion between the adsorbent and adsorbate, the rapid control steps in adsorption, and the factors influencing the adsorption rate, which are closely related to the reaction time. The rate constants in adsorption kinetic models are frequently used to describe the adsorption rate. Frequently used adsorption kinetic models include pseudo primary adsorption kinetic models and pseudo secondary adsorption kinetic models [22].

(1)Pseudo-first-order model adsorption kinetics model.

The differential form of the pseudo-first-order model dynamic model equation is as follows [23]:(5)$$\frac{{\mathrm{dq}}_{t}}{\mathrm{dt}}=k_{1}\left(q_{e}-q_{t}\right)$$

The boundary conditions of the integral are: when t = 0, q_e_ = 0; when t = t, q_t_ = q_t_.

Where q_e_ refers to the adsorption amount when the adsorption equilibrium is reached; q_t_ refers to the adsorption amount at any time during the adsorption process; t is the adsorption time, min; k_1_ is the adsorption rate constant of the pseudo-first-order model adsorption kinetic model [24,25].

(2)Modified pseudo-first-order model dynamics: The modified pseudo-first-order model kinetic model is obtained by transforming the rate constant in the Equation (6). K_1_(min^−1^) is the rate constant of the modified equation of the pseudo-first-order model dynamics model [26].

When t ∈ (0,t), integrate the above equation, and in this range, the adsorption concentration increases from 0 to qt, and the following equation can be obtained:(6)$$\frac{q_{t}}{q_{e}}+\ln(q_{e}-q_{t})=\ln(q_{e})-k_{1}t$$

If the adsorption process follows the modified first-order kinetic model equation, then the rate constant K_1_ and adsorption quantity q_e_ can be obtained by drawing a line with (q_t_ − q_e_) + ln(q_e_ − q_t_) to t [27].

(3)Pseudo-second-order kinetic model [28,29].

The equation of the pseudo-second-order dynamics model is expressed as follows:(7)$$\frac{t}{q_{t}}=\frac{1}{k_{2}q_{e}^{2}}+\frac{1}{q_{e}}t$$

In the above equation, q_e_ is the equilibrium adsorption amount, and k_2_ is the equation constant of the pseudo-second-order kinetic model. Using t/q as a line against t, q_e_ and k_2_ can be found from the slope and intercept of the line on the graph, and {τ = k_2_q_e_^2^} can be found [30,31].

## 3. Results

### 3.1. Effect of Operation Parameters

As shown in Figure 1, the effect of MCS addition and initial pH of the solution on the adsorption of Cu^2+^ and Pb^2+^ was investigated. The experimental procedure was as follows.

In order to adjust the pH of the 200 mg/L Cu^2+^ and Pb^2+^ solutions to 2.00, 3.00, 4.00, 5.00, and 6.00, different concentrations of NaOH and HNO_3_ solutions were used, respectively. First, 0.0040 g, 0.0080 g, 0.0120 g, 0.0160 g, and 0.0240 g of porous calcium silicate adsorbent was added to each of the five groups of six 100 mL centrifuge tubes (30 in total), and 0.0200 g, 0.0240 g of MCS adsorbent and 20 mL of Cu^2+^ solution with a concentration of 200 mg/L, pH = 2.00~6.00 to were added each group, respectively. After being sonicated for two minutes using a high-power CNC ultrasonic cleaner, the samples were shaken for 24 h at 303 K in a water bath. In this experiment, samples were collected after the shaking time had elapsed, the pH of each sample was measured, and the solution was then taken using a 5 mL syringe and filtered through a 0.22 μm pinhole filter. Filtered samples were diluted and volumetrically determined using a stoppered colorimetric tube. By atomic absorption spectrometry, the concentration of Cu^2+^ in the samples was determined in order to determine the removal rate and adsorption capacity of the heavy metal ions Cu^2+^.

It can be seen from Figure 1a that the removal rate of heavy metal ions Cu^2+^ increased with the addition of MCS at pH = 2. The effect of increasing the removal rate was slow when the addition amount was between 0.2 and 1.0 g/L. When the pH = 3, 4, 5, and 6, the removal rate of heavy metal ions Cu^2+^ increased with the addition of 0.2~0.6 g/L. After the addition of 0.6 g/L, the removal rate tended to be stable. As can be seen from Figure 1b, the removal rate of heavy metal ions Pb^2+^ did not change significantly with the addition of porous calcium silicate at pH = 2; at pH = 3, the removal rate of Pb^2+^ increased with the addition of porous calcium silicate up to 0.3 g/L, and then stabilized, and the trend was most obvious between 0.15 and 0.25 g/L. At pH = 4, 5, and 6, the removal rate of Pb^2+^ increased with the addition of porous calcium silicate up to 0.15 g/L, and then stabilized. At pH = 4, 5, and 6, the removal rate of Pb^2+^ increased with the addition of porous calcium silicate until 0.15 g/L and then stabilized. When the pH was greater than 2, the best adsorbents of MCS for Cu^2+^ and Pb^2+^ were 0.6 g/L and 0.15 g/L, respectively, at which time Cu^2+^ and Pb^2+^ in solution were almost completely adsorbed by MCS. It can also be seen that the affinity of MCS for Pb^2+^ was greater than that for Cu^2+^.

In Figure 1c, when pH = 2, the adsorption capacity of heavy metal ion Cu^2+^ increased with the increase of MCS addition, and the adsorption capacity increased slowly when the addition amount was between 0.2 and 1.0 g/L. However, after 1.0, the increase effect had an obvious change; when pH = 3, the adsorption capacity increased with the increase of porous calcium silicate addition amount until 0.6. At pH = 4, 5 and 6, the adsorption capacity increased with the addition of porous calcium silicate up to 0.4 g/L, but decreased with the addition of MCS after 0.4 g/L. In Figure 1d, the adsorption capacity of heavy metal ion Pb^2+^ did not change significantly with the increase of MCS addition at pH = 2; at pH = 3, the adsorption capacity increased with the increase of MCS addition up to 0.2 g/L, and the trend of increase was most obvious between 0.15 and 0.2 g/L, but decreased with the increase of MCS addition after 0.2 g/L. At pH = 3, the adsorption capacity increased with the increase of MCS addition up to 0.2 g/L, but decreased with the increase of MCS addition after 0.2 g/L. At pH = 4 and 5, the adsorption capacity increased with the addition of porous calcium silicate up to 0.15 g/L, but decreased with the addition of MCS after 0.15 g/L. At pH = 6, the adsorption capacity of Pb^2+^ decreased gradually with the addition of MCS. It can be seen from the graph that the adsorption capacity of MCS for Pb^2+^ was greater than that for Cu^2+^.

We conclude that H^+^ in solution inhibits the adsorption of Cu^2+^ by MCS. Despite the fact that the equilibrium pH in solution increased after adsorption equilibrium in Figure 1e,f, this was likely due to the exchange of -OH with heavy metal ions on the surface of MCS during the adsorption process, which allowed -OH to enter the solution and increase the pH value. It also explains why when the pH was low, the H^+^ in solution reacted preferentially with -OH, resulting in a lower removal of heavy metal ions.

### 3.2. Characterization of MCS 

Figure 2 shows the XRD patterns of MCS. As seen in Figure 1a, the peaks of MCS were narrow and mainly consisted of Ca_3_SiO_5_, Ca(OH)_2_, and CaCO_3_ phases. CaCO_3_ was formed by the carbonation of Ca(OH)_2_ that absorbed CO_2_ from the air. The Figure 2b illustrates the XRD pattern of MCS after the adsorption of copper ions. We observed only the peak of Ca_3_SiO_5_, and the peak intensity has decreased. This can be attributed to the chemical adsorption of Cu^2+^ on MCS, which reduced the crystallinity of calcium silicate. The calcium silicate did not completely react and adapted to adsorb on the surface of the MCS, preventing any further chemical adsorption reactions.

As shown in Figure 2c, MCS after the adsorption of lead ions exhibited an XRD pattern. Based on the phase analysis in the Figure 2c, the main components were 2PbCO_3_·Pb(OH)_2_, Pb_3_(CO_2_)_2_ (OH)_2_, and Ca_3_SiO_5_, while the peak of calcium silicate almost disappeared. This indicates that chemical adsorption between MCS and Pb^2+^ was more intense. The peak of calcium silicate at 2θ = 29.43° had a stronger intensity for (a) > (b) > (c), and the adsorption capacity of calcium silicate for Cu^2+^ and Pb^2+^ was the same.

Figure 3a shows the sample exhibits characteristic absorption peaks at approximately 3440, 1630, 1440, 966, 664, and 471 cm^−1^. These peaks corresponded to functional groups as follows: the peak at approximately 3440 cm^−1^ corresponded to the asymmetric stretching vibration of water molecules and surface -OH groups. A peak at around 1630 cm^−1^ was caused by the vibration of adsorbed water molecules and surface -OH, indicating the presence of abundant reaction groups -OH on the sample surface. The peak around 1440 cm^−1^ was attributed to the calcium silicate peak, while 966 cm^−1^ was attributed to the symmetric stretching vibration of Si-O-Si. The characteristic peaks at 664 cm^−1^ and 471 cm^−1^ were caused by the deformation of Si-O-Si. From Figure 3b,c, it can be observed that the peaks of MCS FT-IR at 966 cm^−1^, 664 cm^−1^, and 471 cm^−1^ disappeared after the adsorption of copper ions and lead ions. Chemical adsorption resulted in the fracture of Si-O-Si and the disappearance of the characteristic peak due to chemical adsorption [32].

We examined the morphology of MCS before and after the adsorption of copper and lead heavy metal ions using scanning electron microscopy. The adsorbed MCS was analyzed using a scanning electron microscope in combination with an energy-dispersive X-ray spectrometer, as shown in Figure 4. The SEM image of MCS is shown in Figure 4a, from which it can be seen that MCS was fluffy and porous, which provided a large number of adsorption sites for the adsorption of heavy metals. Figure 4b shows MCS after the adsorption of Cu^2+^. It can be seen that the surface of MCS was covered by a layer of particles, and the thickness of MCS increased, indicating that copper ions were adsorbed on the surface and pores of MCS, making the porous calcium silicate no longer fluffy. This is confirmed by the energy-spectrum distribution map in Figure 4f. Figure 4c shows the morphology of MCS after adsorption of lead ions. The morphology of MCS changed from a fluffy porous state to a dense flaky state, indicating that the adsorption of Pb^2+^ is not simply a physical adsorption process [33].

In Figure 5, the nitrogen adsorption-desorption isotherms and pore size distribution curves are shown before and after the adsorption of copper and lead heavy metal ions by MCS. The specific surface area and the pore size of the samples were significantly reduced as a result of the adsorption of copper and lead ions by MCS. The specific surface area and pore size of MCS were calculated by the BET method and Barrett–Joyner–Halend method, which were 69.92 m^2^/g and 38.6 nm, respectively, while the specific surface area and average pore size of MCS after adsorption of copper and lead ions were 42.45 m^2^/g, 30.4 nm, 20.49 m^2^/g, and 28.8 nm, respectively. The adsorption of older heavy gold ions on the surface and in the pores of MCS, as shown in Figure 3, resulted in a reduction of both the specific surface area and pore size.

### 3.3. Isothermal Adsorption Mode

The adsorption isotherms of Cu^2+^ and Pb^2+^ (293 K, 303 K and 313 K) at different temperatures are shown in Figure 6. It was found that the adsorption capacity of MCS for Cu^2+^ increased continuously with the increase of initial concentration, and that it increased rapidly when the initial concentration of Cu^2+^ was below 200 mg/L. The Cu^2+^ concentration continued to increase, but the q_e_ increased slowly. When the initial concentration of Pb^2+^ was lower than 20 mg/L, the adsorption capacity of MCS increased rapidly with the increase of the initial concentration of Pb^2+^, and when the concentration of Pb^2+^ was greater than 200 mg/L, the adsorption capacity almost ceased to change. Meanwhile, it is evident from Figure 5 that the adsorption capacity q_e_ of MCS for heavy metal ions Cu^2+^ and Pb^2+^ increased with the increase of temperature. Therefore, the increase in temperature is beneficial for the adsorption of heavy metal ions Cu^2+^ and Pb^2+^ on MCS.

Table 1 and Table 2 show the fitted parameters of Langmuir and Freundlich for MCS to heavy metal ions Cu^2+^ and Pb^2+^, respectively. The correlation coefficient (R^2^) fitted by the Freundlich model was higher than that of Langmuir. This result indicates that the description of the adsorption of heavy metal ions by MCS is more consistent with the Freundlich model, suggesting that the adsorption process is not limited to monolayer adsorption.

Based on the Freundlich model, the maximum adsorption capacity q_e_ of MCS for Cu^2+^ and Pb^2+^ was 561.885 mg/g and 1921.506 mg/g, respectively. This indicates that the affinity of MCS for lead ions was much greater than that for copper ions.

### 3.4. Adsorption Kinetics Study

The effect of mesoporous calcium polyacid on the adsorption capacity at different adsorption times for heavy metal ions with initial concentrations of 200 mg/L, 400 mg/L and 600 mg/L; the variation curves with contact time; the fitted first-order and second-order fitted lines; and the fitted parameters of the obtained copper and lead adsorption capacities are shown in Figure 7. From Figure 7a, it can be seen that the adsorption times for MCS to reach adsorption equilibrium for Cu^2+^ initial concentrations of 200 mg/L, 400 mg/L, and 600 mg/L were 30 min, 90 min, and 360 min, respectively.

As shown in Figure 7b, the adsorption time for MCS to reach the adsorption equilibrium for Pb^2+^ initial concentrations of 200 mg/L, 400 mg/L, and 600 mg/L, respectively, was 360 min. There was a linear relationship between the adsorption capacities of Cu^2+^ and Pb^2+^ in the same period of time: 600 mg/L > 400 mg/L > 200 mg/L. This indicates that the adsorption capacity of MCS for Cu^2+^ and Pb^2+^ at the same adsorption time is related to the concentration of Cu^2+^ and Pb^2+^ in the original solution. Pb^2+^ is more easily adsorbed by MCS than Cu^2+^, and this is due to the high surface area of the adsorbent and the distinct adsorption sites with Cu^2+^ as opposed to Pb^2+^, which form complexes more easily.

The correlation coefficients for the kinetics of Pb^2+^ and Cu^2+^ adsorption by MCS are shown in Table 3, and the fit is better when the heavy metal ion concentration is 200 mg/L. Therefore, the corresponding parameters are shown in Table 3. The fit coefficients for the pseudo first-order and pseudo second-order kinetic models indicate that the pseudo second-order model is a better fit and is the best model for describing the kinetics of Cu and Pb ion adsorption. They also indicate that the mesoporous calcium silicate for Cu^2+^ and Pb^2+^ adsorption is mainly chemisorption. During the adsorption process, heavy metal ions form chemical bonds by sharing or exchanging electrons and attach to the MCS surface, i.e., surface complexation.

### 3.5. Effect of Ionic Strength on Adsorbent

To study the effect of solution ionic strength on the adsorption process, this study used sodium nitrate as the background electrolyte to investigate the adsorption performance of MCS adsorbent on Pb^2+^ and Cu^2+^ in different concentrations of NaNO_3_ solution. In this experiment, NaNO_3_ was used at concentrations of 0.001 mol/L, 0.01 mol/L, 0.05 mol/L, 0.2 mol/L, and 0.2 mol/L, respectively. Then, the MCS adsorbent was added to study the effect on the removal rate of Pb^2+^ and Cu^2+^.

As shown in Figure 8, Na^+^ has no effect on the adsorption of Pb^2+^ and Cu^2+^ by MCS; thus, it is clear that the use of sodium nitrate as a background electrolyte does not affect the adsorption performance of MCS. The fact that MCS does not specifically adsorb sodium and nitrate ions does not affect the results of the experiment, i.e., sodium nitrate is an irrelevant electrolyte.

### 3.6. Elution, Recycling, and Recovery of Adsorbed Heavy Metals

Table 4 shows the elution and recovery rates of adsorbed Cu^2+^ and Pb^2+^ on MCS. Using 0.1 mol hydrochloric acid solution as the eluent, the experimental results are shown in Table 4. The elution rates of MCS-Cu and MCS-Pb using 0.1 mol hydrochloric acid as the eluent were both high and almost completely removed. This is because the lower pH conditions resulted in the dissolution of the MCS, allowing the heavy metals adsorbed on the MCS to dissolve in solution. Meanwhile, the heavy metal ions in solution were recovered almost completely using sodium hydroxide as a precipitant.

## 4. Discussion

Although China has abundant bauxite reserves, it is mainly bauxite of diaspore and kaolinite type, which is generally characterized by high aluminum, high silicon, a low aluminum-silicon ratio, high impurities, and miscellaneous components. Moreover, the annual consumption of resources is also enormous. With the increasing demand for bauxite and the depletion of domestic high-grade bauxite resources, the key to solving the strategic demand for bauxite is to improve the Al-Si ratio of low-grade bauxite and make it suitable for Bayer process production by pre-desilication. However, a large amount of desilication lye will be produced in the desilication process, and a large amount of slag will be produced during the desilication process, which will increase the production of red mud, which not only increases the cost but also causes a significant burden on the environment.

According to the above experimental conclusions, the synthesis of MCS from low-grade bauxite desilication liquid has an excellent adsorption effect on copper and lead ions. Using low-grade bauxite desilication liquid to prepare MCS can reduce the increase of solid waste produced during the mother liquor cycle, which is helpful to increase the economic benefit of bauxite. MCS is used to adsorb heavy metals in waste liquid to achieve the effect of abolishing, which provides a new idea for improving the process of the bauxite industry. Therefore, preparing MCS using low-grade bauxite desilication liquid has a good application prospect, which is more green and offers better environmental protection.

## 5. Conclusions

In this experiment, MCS was prepared by the hydrothermal reaction of added lime with a low-grade bauxite desilication solution and used as an adsorbent to adsorb copper and lead ions from wastewater. The main findings are as follows.

The MCS had high adsorption capacity for Cu^2+^ and Pb^2+^ when the adsorption solution was pH = 4~6, with Pb^2+^ (1921.506 mg/g) > Cu^2+^ (561.885 mg/g). It was found that the lower pH inhibited the adsorption of heavy metal ions from the solution by the adsorbent. The final adsorption conditions for Cu^2+^ and Pb^2+^ were pH = 4, with an adsorption temperature of 303 K. The adsorption rates of Cu^2+^ and Pb^2+^ after 24 h were 99.51% and 99.74%, respectively.The kinetic adsorption of Cu and Pb ions by MCS was calculated to be more in line with the quasi-secondary kinetic model. The adsorption capacity of Cu^2+^ and Pb^2+^ was related to the initial solution concentration, which was 600 mg/L > 400 mg/L > 200 mg/L. The results showed that the adsorption of Cu^2+^ and Pb^2+^ by MCS was mainly chemisorption. During the adsorption process, heavy metal ions form chemical bonds by sharing or exchanging electrons and attach to the MCS surface.Based on the calculations of the isothermal adsorption models for Cu^2+^ and Pb^2+^ by MCS, it was proposed that the adsorption of Cu^2+^ and Pb^2+^ by MCS was not limited to monolayer adsorption. It has been demonstrated that an increase in the adsorption temperature promotes the adsorption of Cu^2+^ and Pb^2+^ from MCS with a maximum adsorption capacity of q_m_(Pb) > q_m_(Cu). According to the Freundlich isotherm, the *n* value for Pb is higher than that for Cu, indicating that the adsorption of Pb^2+^ by MCS is greater than that of Cu^2+^.By adjusting the background electrolyte NaNO_3_ concentration, it was found that the background electrolyte had no effect on the over-sorption process during the adsorption of Cu^2+^ and Pb^2+^ by MCS.

## Figures and Tables

**Figure 1 materials-16-03506-f001:**
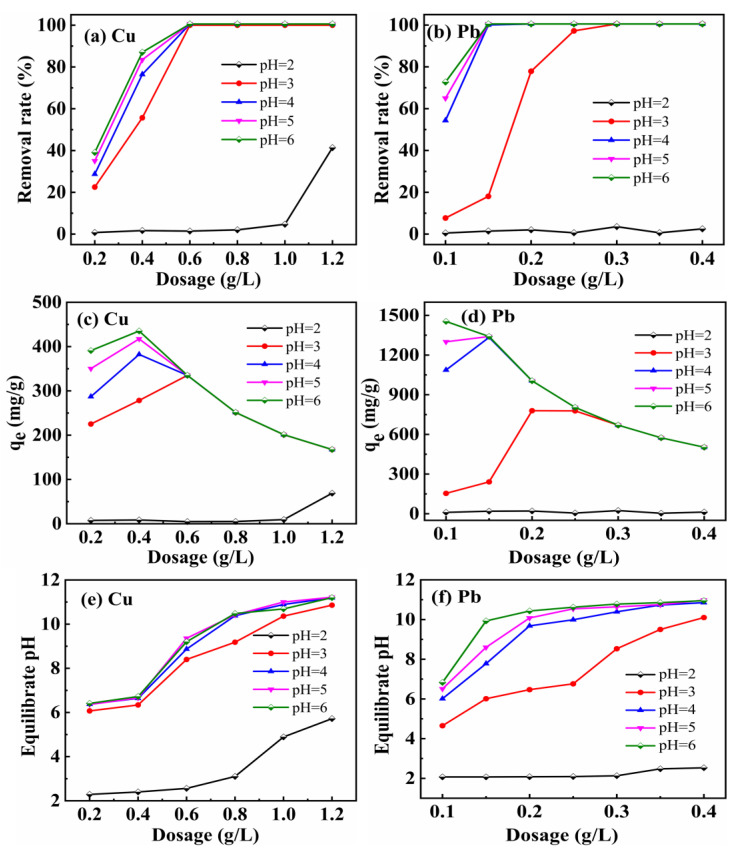
Effect of initial solution pH and adsorbent addition amount on adsorption of Cu^2+^, Pb^2+^: Cu^2+^ removal rate (**a**), Cu^2+^ adsorption capacity (**c**); Cu^2+^ equilibrate pH (**e**); Pb^2+^ removal rate (**b**), Pb^2+^ adsorption capacity (**d**); Pb^2+^ equilibrate pH (**f**).

**Figure 2 materials-16-03506-f002:**
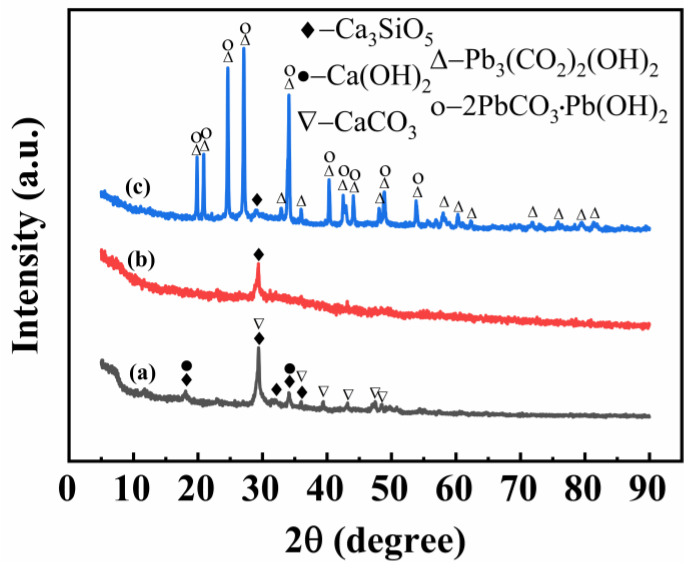
X-ray diffraction patterns of MCS (**a**) and MCS loaded with Cu^2+^ (**b**), Ni^2+^ (**c**).

**Figure 3 materials-16-03506-f003:**
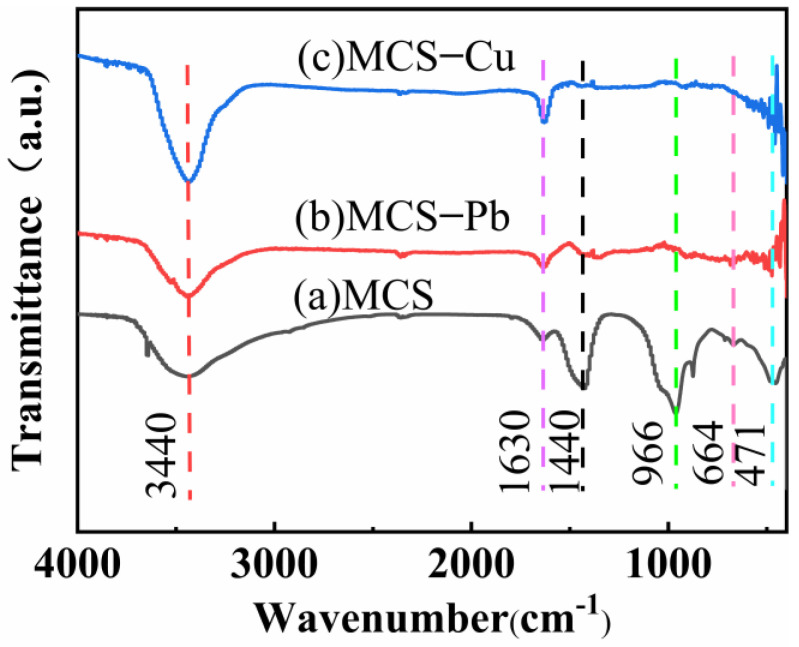
Fourier transform infrared spectroscopy spectra of MCS (**a**) and MCS loaded with Pb^2+^ (MCS−Pb) (**b**), Cu^2+^ (MCS−Cu) (**c**).

**Figure 4 materials-16-03506-f004:**
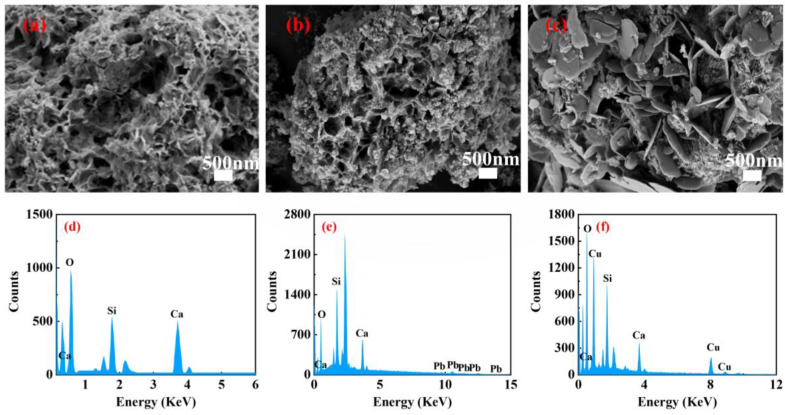
Scanning electron microscopy images of MCS (**a**) and MCS loaded with Cu^2+^ (**b**), Pb^2+^ (**c**). Energy spectrum analysis diagram (**d**) of MCS: MCS−Pb (**e**), MCS−Cu (**f**).

**Figure 5 materials-16-03506-f005:**
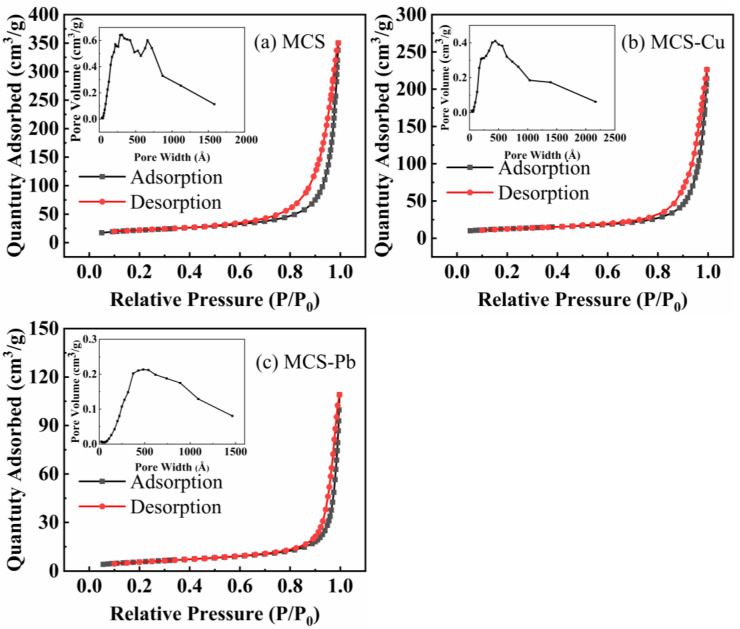
Nitrogen adsorption-desorption isotherms and pore size distribution curves of MCS (**a**) and MCS loaded with Cu^2+^ (MCS−Cu) (**b**), Pb^2+^ (MCS−Pb) (**c**).

**Figure 6 materials-16-03506-f006:**
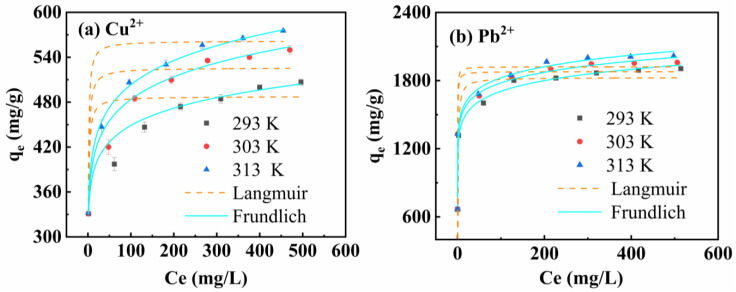
Adsorption isotherms of MCS for Cu^2+^ (**a**), Pb^2+^ (**b**).

**Figure 7 materials-16-03506-f007:**
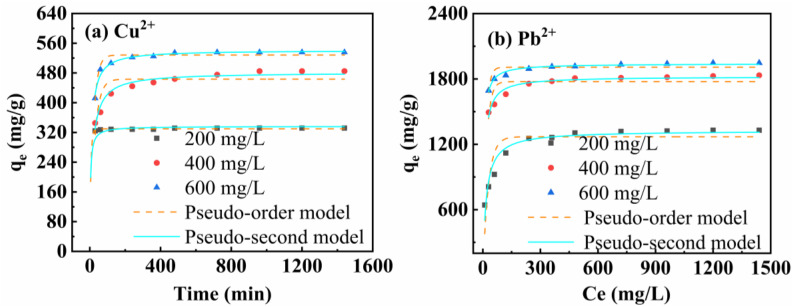
Change in adsorption amount (qt) with time (t) at different Initial concentrations: Cu^2+^ (**a**), Pb^2+^ (**b**).

**Figure 8 materials-16-03506-f008:**
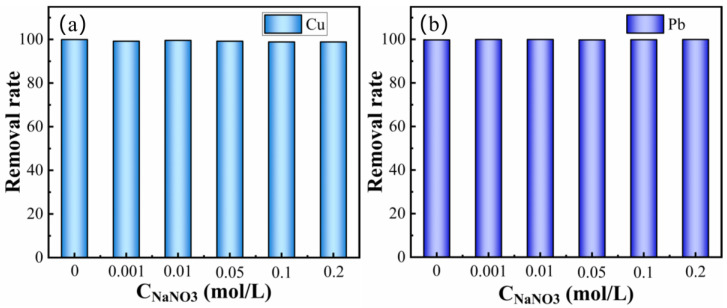
Effect of ionic strength on adsorption of heavy metal ions by MCS: Pb^2+^ (**a**), Cu^2+^ (**b**).

**Table 1 materials-16-03506-t001:** Parameters of the Cu^2+^ isotherm model for MCS adsorption.

Langmuir Equation				
Name	T (K)	q_m_ (mg/g)	K_L_ (L/mg)	R^2^
MCS-Cu	293	487.622	1.407	0.975
	303	525.947	1.278	0.982
	313	561.885	1.485	0.987
Freundlich equation				
Name	T (K)	K_F_(mg/g)(L/mg)^1/*n*^	*n*	R^2^
MCS−Cu	293	321.234	13.774	0.997
	303	322.938	11.407	0.999
	313	332.606	11.123	1.000

**Table 2 materials-16-03506-t002:** Parameters of the Pb^2+^ isotherm model for MCS adsorption.

Langmuir Equation				
Name	T (K)	q_m_ (mg/g)	K_L_ (L/mg)	R^2^
MCS−Pb	293	1827.013	1.071	0.617
	303	1879.031	4.723	0.657
	313	1921.506	6.643	0.676
Freundlich equation				
Name	T (K)	K_L_ (mg/g)(L/mg)^1/*n*^	*n*	R^2^
MCS−Pb	293	1194.354	12.918	0.993
	303	1269.236	13.612	0.981
	313	1293.008	13.367	0.978

**Table 3 materials-16-03506-t003:** Kinetic parameters of adsorption of Cu^2+^ and Pb^2+^.

Model	Parameters	Cu^2+^			Pb^2+^		
		200 mg/L	400 mg/L	600 mg/L	200 mg/L	400 mg/L	600 mg/L
Pseudo-first-order model	R^2^	0.956	0.731	0.935	0.766	0.640	0.721
	k_1_	169 × 10^−3^	37.8 × 10^−3^	48.8 × 10^−3^	35.6 × 10^−3^	55.1 × 10^−3^	69.9 × 10^−3^
	q_e_ (cal)	329.725	463.319	528.238	1269.23	1775.172	1907.849
Pseudo-second-order model	R^2^	0.997	0.951	0.974	0.932	0.938	0.968
	k_2_	0.858 × 10^−3^	0.148 × 10^−3^	0.216 × 10^−3^	0.046 × 10^−3^	0.071 × 10^−3^	0.112 × 10^−3^
	q_e_ (cal)	336.386	481.484	541.352	1325.538	1822.337	1940.063

**Table 4 materials-16-03506-t004:** Elution and recovery of Cu^2+^ and Pb^2+^ from the adsorbed MCS.

Items	Cu^2+^	Pb^2+^
Elution ration (%)	99.87	99.95
Recovery ratio (%)	99.64	99.78

## Data Availability

All data supporting this study’s findings are included within the article.

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
