# Peer review of "Preparation of MCS from Low-Grade Bauxite Desilication Lye and Adsorption of Heavy Metals"

_materials, 2023, doi:10.3390/ma16093506_

Round 1

Reviewer 1 Report

Dear authors,

The subject of this paper is interesting and has a lot of potential

for application in environmental protection your manuscript needs

a lot of improvement and correction.

I advise you to carefully review the chemical formulas of the compounds you use.

When using NaNO3 as background electrolyte by adjusting NaOH and HNO3 solutions of different concentrations - you will vary not just the ionic strength but the pH as well.

Detail comments:

1.     Row 27 “The most common way to treat heavy metals in solution is to use adsorbents to ad-27 sorb heavy metals in solution.” – it is one of the methods, but I am not sure that the statement “the most common” is accurate.

2.     Row 32 - TiO  - incorrect formulae

3.     Row 59 - (H2NO3, purity ≥99), Copper nitrate trihydrate (CuNO3·3H2O, purity ≥99), lead nitrate hexahydrate (PbNO3·6H2O) – incorrect formulae

4.     Row 76 – “… different NaOH and HNO3 solution concentrations were used to adjust the pH of…” – it would be more appropriate to use buffer solutions for fixed pH of the media, especially if you are evaluating its effect. The buffers are used to maintain the pH of the media throughout the whole reaction not just in the beginning.

5.     Row 85 – “… were substituted into (1-1) and (1-2) to calculate…” – the equations they are referring to are labeled as (1) and (2)

6.     Row 106 – “…and void volume…” – “pore” is the appropriate term.  

7.     Figure 1 is of poor quality.

8.     In Figure 2 – Ca(CO)3  - incorrect

9.     Figure 5 is of poor quality.

10.  Row 363 – “The concentration of NaNO3 was adjusted by using NaOH and HNO3 solutions of different concentrations.” – this way pH of the media is also changed, so it is no accurate to state that this is just effect of ionic strength.

11.  Figure 7 – the description does not correspond to the graphics.

12.  The English should be improved as well.

Reviewer 2 Report

This manuscript cannot be accepted in present form for publication. The authors have to reconsider it and improve the presentation.

1. Chemical formulae for nitric acid, copper nitrate and lead nitrate are wrong.

2. English is very poor and makes some phrase difficult to understanding.

3. There are some orthographic mistakes.

4. The rate constant is k1 not K1.

5. The equation for quasi-first kinetic model is (7) not (1-7) in the text.

6. In section 2.3 the authors tell that Ci=100 mg/L but in section 3.1 Ci=200 mg/L. What is correct?

7. The authors have to correct the method for determination metals on section 2.3. This is not carried out by a simple spectrometer. In section 3.1 they written that the concentration was determined by flame atomic absorption spectrometry.

8. Figure 6a represent in fact figs 6b+6c+6d. Please correct.

9. Fig 6 is the same with fig 7.

10. The explanations for figs. 6,7,8 and 9 are wrong.

11. Why this sorbent is better than others?

12. Can be re-used this sorbent?

13. What is selectivity of these sorbent?

14. Why this sorbent retain better Pb(II) than Cu(II)?

15. In references section some title of the journals are abbreviated and some not.

Reviewer 3 Report

The article deals with investigation of new adsorbent based on mesoporous calcium silicate for lead and copper removal. The manuscript addresses an issue that is very important and attractive. Results in this manuscript are worthy; however, major manuscript issues are: lack of clarity and the paper is too descriptive (the results are described but not explained). I do believe there is valuable data to be published from this research, but changes should be made.

Title: Preparation of mesoporous calcium silicate from low-grade bauxite desilication lye and adsorption of heavy metals

 Article Type: Original paper

 Materials and methods

There is lack of information regarding all equipment used in experiments (pH-meter, SEM, FTIR, XRD, etc.).

Line between 69 and 73 (subsection 2.2) should be transfer in to Section 3 - Results. Authors should introduce new subsection “Characterisation of adsorbents” and to describe methodology and equipment used for characterisation (XRD, SEM, FTIR).

Line 74 (subsection 2.3): this subsection should be written more clearly. In order to examine the sorption process it is necessary to determine the effects of different operating parameters such as pH, sorbent concentration, metal concentration, contact time and temperature. The batch adsorptions studies identify the optimum parameters affecting the efficiency of metal removal by adsorbent. All those experiments should be done within same conditions, varying just one of parameters, and afterwards according to obtained optimal parameters authors exam isotherm and kinetic models.

Line 82:  “The Cu2+ and Pb2+ concentrations of the samples were determined by spectrophotometer”. Correct spectrophotometer into Atomic Absorption Spectrometry. Define equipment.

Results and discussion

Line 164: please change title of subsection 3.1 into “Effect of operation parameters” (it is not that only influence of solution pH was described in this subsection).

Line 167: why initial metal concentration is 200 mg/L when in subsection 2.3 authors said that initial metal concentration was 100 mg/L?

Figure 1: Instead of describing the Figure 1, authors should include some explanation about behavior of MSC at different pH value of solution. Is a material pH value effect its behavior in solution with different pH value? What is pH value of MSC (you should determine the pH suspension: mass of 0.2 g of material suspend in 30 cm3 distilled water and left in a closed container to avoid contact with air. The suspension stirs occasionally for 72 h and afterward measure pH value. Measured value is pHsus.). Also authors should have measured pH final (in supernatant - after sorption) to see if any change occurred, and then to try to give explanation.

Finally, why authors chose the 0.8 g/L as an optimum dosage?

Figure 2: evidently MSC has a greater affinity toward lead ions (I believe that is the reason why authors presented effect of dosage onto x axis only in the range from 0.1 to 0.4 g/L), but why authors choose 0.8 g/L as an optimum value?

Lines 220-223: observations are not in accordance with subsection 2.3 (why authors choose 0.8 g/L as an optimum value?).  

Line 235: PbCO3!Pb(OH)2 – typos.

Figure 4: MSC energy spectrum analysis diagram is missing.

Line 286: In Figure 6 and 7 non-linear models are presented not linear. Put operation parameters in Figures (6,7,8 and 9) capture.

Figure 6 and 7 are the same! Please check!

Line 292: “It can be seen from Figure 6 and Figure 7 that the adsorption capacity of MCS for heavy metal ions Cu2+ and Pb2+ increases with the concentration of heavy metal ions”. I am not shore what is the meaning of this observation: on x axis author should have put residual (final) concentration of metal in solution after process of sorption.

Line 295: put it in subscript: Cu2+ and Pb2+

Line 311: Table 1: correct name into MSC-Cu. Same for Table 2 (MSC-Pb).

Line 315: where can one find results of RL?

Line 318: Table 2: please check the values of qm (it’s too high, almost 2g/g).

Line 319: uniform the name of kinetic models:  is it pseudo or quasi?

Line 323: “the adsorption rate is more than 99 %” – Figure 8 and 9 presents adsorption capacity not Removal rate or Adsorption efficiency.

Line 327: Figure 6?

Explanation of Figure 8 and 9 are descriptive again. Table with kinetic parameters obtained by fitting with kinetic models is missing.

Comparison with the results obtained from other researchers who investigated similar materials is missing.

Round 2

Reviewer 1 Report

Dear authors,

There are still several issues that bother me:

1) Why is the text highlighted?

2) row 357  -  "Na+ has no effect on the adsorption of Pb2+ and Cu2+ by MSC. Therefore, it is feasible to use HNO3" In the end what substance did you use as a background electrolyte -NaNO3 or HNO3, because if the latter - you will also change pH not just the ionic strength, which may lead to incorrect conclusions.

3) In Figure 1 it would be better if you level/compare the corresponding trends for copper and lead ions - (a) and (d); (b) and (e); (c) and (f).

4) There is difference in the size and font throughout the text of the manuscript - please make sure to make it uniform.

Kind regards,

3)

Reviewer 2 Report

The response for 14 requirement is in Chinese!!!!

The authors have to carry out the desorption of the metals from the adsorbent and re-use in some cycles adsorption/desorption in order to highlight the reusability degree of this adsorbent.

Reviewer 3 Report

Dear authors:

In Figure 1., x axis still have different values for sorbent dosage: Fig 1a, 1b and 1c are in the range from 0.2 to 1.2 g/L and Fig 1d, 1e and 1f are in the range from 0.1 to 0.4 g/L. Make it uniform.

 In Figure 4., MSC energy spectrum analysis diagram is still missing. And I didn’t understand authors response:

Response:尊敬的审稿人,样品已经再送样中,但由于最近毕业季,实验样品仍在排队中,等实验样品检测了我再与编辑联系补上这个数据可以吗?

 Table 3., should have all kinetic parameters not only R2.
